# Improved Oral Health and Adaptation to Treatment in Dogs Using Manual or Ultrasonic Toothbrush or Textile of Nylon or Microfiber for Active Dental Home Care

**DOI:** 10.3390/ani11092481

**Published:** 2021-08-24

**Authors:** Lena Olsén, Anna Brissman, Sara Wiman, Fanny Eriksson, Camilla Kaj, Karolina Brunius Enlund

**Affiliations:** Department of Clinical Sciences, Swedish University of Agricultural Sciences, SE-750 07 Uppsala, Sweden; aabn0004@stud.slu.se (A.B.); sawi0002@stud.slu.se (S.W.); fyer0001@stud.slu.se (F.E.); caka0004@stud.slu.se (C.K.); karolina.enlund@slu.se (K.B.E.)

**Keywords:** FAS, gingivitis, plaque, calculus, periodontitis, tartar, finger cloth, ultrasonic toothbrush Handy Brush, Accesia Softy Swipe, Emmi pet

## Abstract

**Simple Summary:**

Poor oral health is a common problem in dogs and can cause them pain and impaired welfare. In this study, four different devices used for active dental home care were examined: two toothbrushes, manual and ultrasonic; and two textile products, a nylon glove and a microfiber finger cloth. These pieces of equipment were evaluated for efficacy against gingivitis, plaque, and tartar, using twenty-one beagle dogs. The dogs had their teeth cleaned once a day during five weeks. Each dog received two treatments (toothbrushes or textiles) but on separate sides of the mouth, right and left, respectively. The dogs’ oral health was assessed before and after the five-week period, by a veterinarian not aware of treatment. Each device improved the oral health in the dogs, reducing plaque and gingivitis. Furthermore, the amount of calculus also decreased in some individuals. The study also assessed the dogs’ stress, fear and anxiety levels during treatment using a (fear, anxiety, stress (FAS) protocol. The assessment of the stress levels in the dogs showed that adaptation occurs in the dogs subjected to daily dental care. These results can be used by veterinary staff who want to inspire and encourage dog owners to start brushing their dog’s teeth.

**Abstract:**

Poor oral health is a common problem in dogs causing pain and impaired welfare. The gold standard to maintain good oral health and prevent periodontitis is daily tooth brushing. However, many dog owners find it difficult, and dogs often show aversive behavior. There are several different tools available, and in this intra-subject study, four different active dental home care treatment options were examined: manual and ultrasonic toothbrush, a nylon glove, and a microfiber finger cloth. The treatments were evaluated for efficacy against gingivitis, plaque, and calculus in beagle dogs. Each dog received two treatments on the buccal sides of the teeth once daily, either two types of toothbrushes (N = 10) or two types of textiles (N = 11), on separate sides of the mouth. The dogs’ teeth were assessed before and after the five-week treatment period, with the assessor blinded by additional, similar dogs (N = 7). The study also assessed the dogs’ stress levels during treatment using a fear, anxiety, stress (FAS) protocol. All four treatments improved dog oral health, reducing gingivitis and plaque. Furthermore, the amount of calculus also decreased in some individuals. Adaptation to dental care occurred in the dogs over time. This information can be used to encourage dog owners to brush their dogs’ teeth.

## 1. Introduction

Dental problems are common in dogs and periodontal disease is found in 80% of dogs over 3 years [1,2,3], and higher in specific breeds, e.g., Yorkshire terriers, where 98% had at least one tooth or aspect with early periodontitis at age 37 weeks [4]. The prevalence of periodontal disease increases with lower body weight and age [1,5]. Periodontal disease is an inflammation affecting the tooth’s supporting tissues, i.e., the gingiva, periodontal ligament, cementum, and alveolar bone [6]. It is initiated by dental plaque, a biofilm consisting mainly of bacteria, which covers the teeth in the absence of tooth brushing. Dental calculus is the mineralized plaque that is not in itself pathological but facilitates the adhesion of dental plaque. Periodontal disease is generally divided into gingivitis, which is reversible with dental home care, graduating to periodontitis [7] which is generally considered irreversible [8]. During the early stage of gingivitis, only the soft tissue is affected, whereas, when it progresses to periodontitis, the bone structures supporting the teeth will also be compromised. Besides possibly leading to discomfort and local complications, associations have been shown between periodontal disease and renal, hepatic, and cardiac disease [9,10,11,12]. Professional dental cleaning under anesthesia without consecutive dental home care is mainly cosmetic and does not protect against periodontal disease [13]. Thus, daily tooth brushing is the gold standard for dental home care and is of uttermost importance to prevent future dental problems [13]. However, the degree of performed tooth brushing in dogs is low, with only 4% of dog owners brushing their dog’s teeth daily [14]. Many dog owners experience difficulties when inspecting and caring for their dog’s teeth [15]. In addition, dental disease often goes unnoticed by owners since dogs often do not show signs of dental discomfort [13]. Before starting to brush an adult dog’s teeth, a thorough dental cleaning and examination under anesthesia is recommended to exclude painful conditions that may obstruct effective training of dental homecare in the dog [13].

There are several different products on the market for removing dental plaque in the gum line in dogs. Textile products are often recommended as an initial means to accustom the dog to dental cleaning, or as an alternative if the dog does not accept the toothbrush. To the authors’ knowledge, there are no previous studies evaluating the effect of textiles for dental cleaning on animals. The ultrasonic tooth brushing technology aims to eliminate the need for mechanical brushing, by breaking down the bacterial chains and pathogens present in the mouth, without pressure or movement. The device generates and transfers ultrasound waves, ranging from 20 kHz up to 10 MHz, depending on the manufacturer [16]. The waves are transferred to a fluid, which leads to the formation of quickly collapsing gas bubbles, resulting in mechanical forces on the tooth surface. It differs from the ultrasonic scaler used by the veterinarian or dentist, where vibrations are transmitted via a metallic tip [16]. There are, to the authors’ knowledge, no scientifically reviewed studies conducted for ultrasonic toothbrushes in dogs. However, the method is well known in human dentistry and studies show an equivalent cleaning effect to electric toothbrushes [16,17].

Besides effectively cleaning the dog’s teeth, the selected method for dental home care should preferably be user-friendly and not be perceived as negative for the dog to minimize stress and discomfort during the procedure. Dogs may find tooth brushing uncomfortable and stressful. Training of the dog to accept brushing should ideally be gradual and include positive reinforcement (treats, praise) so as not to negatively affect the owner–animal bond [13]. To assess early physiological changes (e.g., tachypnea, dilated pupils), and behavioral changes (e.g., licking, altered posture), there are assessment protocols that can be used, one of which is the fear, anxiety, and stress (FAS) scale [18].

The aim of the study was to examine the effect on dental health from four different methods of dental home care in dogs, two types of toothbrushes (manual and ultrasonic), and two types of textiles (a nylon glove and a microfiber finger cloth). An additional aim was to explore whether dogs habituate to the procedure during 35 days of daily dental cleaning.

## 2. Material and Method

### 2.1. Study Population

The toothbrush study (manual and ultrasound) included ten female beagle dogs, and the textile study (nylon and microfiber) included three males and eight female beagle dogs aged 2–12 years (Table 1). In order to allow better blinding, seven additional beagle dogs were used in the evaluations (Table 2).

All dogs were housed and used as teaching dogs at the Swedish University of Agricultural Sciences (SLU) in Uppsala. All dogs lived under the same conditions in groups of 3–6 individuals separated by sex in separate indoor pens with an associated outdoor yard. They were fed dry food twice a day, had free access to water, and had free access to ox-skin chewing bones. The feed given was Hills Vet Essentials Adult Medium or Hills Science Plan Mature Adult 7+ with chicken, according to age. As training rewards, all dogs received the same sort and similar amount of treats, to ensure the same conditions for all participating dogs in the study.

The dogs and facilities used in the study was approved by the Ethics Committee for Animal Experimentation, Uppsala, Sweden Approval No: Dnr 5.2.18-7454/15, User permit: Dnr 5.2.18-2636/15, Teaching permit: Dnr 5.8.18-15533/2018. The study does not need additional approval as no invasive experiments were performed.

## 3. Material

### 3.1. Dental Cleaning Procedures

For 35 days, dental cleaning was performed once daily. The canine teeth and premolars in the upper and lower jaw, and the molars in the upper jaw were treated for one minute on the buccal side.

Eleven dogs—three males and eight females—were treated with two different dental cleaning textiles, nylon vs. microfiber, one on each side of the mouth. Treatment sides were allocated by randomization. The remaining ten dogs—ten females—were similarly treated with two different combinations of toothbrush and toothpaste, one on each side of the mouth. Treatment sides were allocated by randomization. The devices were used to evaluate efficiency of standard recommended methods likely to be used by owners at home.

The effectiveness of each treatment was evaluated by comparing the oral health before the 35 days of treatment with the oral health upon completion of the study. Each dog thus served as its own control. To lessen the risk of biased evaluation, the veterinarian performing the evaluations was presented with 28 different dogs: the 21 participating in the study, and an additional 7 non-participant beagles of similar ages (Table 1 and Table 2) as a means of blinding. The four cleaning treatments evaluated were as follows:

#### 3.1.1. Ultrasonic toothbrushing

This involved using a Emmipet Ultrasonic toothbrush (Emmi-pet, Emmi-ultrasonic, Germany) with separate toothbrush heads for each dog, model A2 (Emmi-pet, Emmi-ultrasonic, Mörfelden-Walldorf, Germany) and a special toothpaste (Emmi-pet, Emmi—ultrasonic, Mörfelden-Walldorf, Germany).

The ultrasonic toothbrush was placed on the maxillary and mandibular canine tooth, the first three premolars, and the fourth premolar, respectively, and held in each place for ten seconds. According to the manufacturer’s instructions, the toothbrush treats up to three teeth at a time, with a recommended treatment time for each tooth of 5–10 s.

#### 3.1.2. Manual toothbrushing

This involved using a Braun Electric toothbrush (Braun, Kronberg im Taunus, Germany) with a separate toothbrush head for each dog (Braun, Kronberg im Taunus Germany) and dog toothpaste (Petosan, Haukeland, Norway).

Initially, the study was designed for the use of an electric toothbrush with oscillating cleaning action. However, during the initial handling training during the week before the start of the treatments, several dogs showed a higher FAS level than the predetermined breaking point (clear FAS 4) when approaching them with the toothbrush switched on. Therefore, the brushing was instead performed as manual tooth brushing using the preselected electric toothbrush but switched off. It was used for 30 s in the upper jaw and 30 s in the lower jaw on the same teeth (but on the opposite side) as for the ultrasonic tooth brushing.

#### 3.1.3. Nylon textile glove tooth cleaning (“nylon textile”)

This involved using a Handy Brush glove with five fingers made of a heavy fabric of double-spun nylon thread (Handy Brush, DogaNova, Kungsängen, Sweden) [19], which were individual and labelled for each dog.

Dental cleaning with textile was performed according to the manufacturer’s instructions, the glove was rubbed over each tooth, and the gingiva was avoided. The dental cleaning was performed without toothpaste during 1 min.

#### 3.1.4. Microfiber finger cloth tooth cleaning (“microfiber”)

This involved using an Accesia Softy Swipe finger cloth in soft microfiber (Accesia [20]), individual and labelled for each dog.

Dental cleaning with textile was performed according to the manufacturer’s instructions: the finger cloth was rubbed over the gingiva and tooth for 1 min. No toothpaste was used.

For dental disclosing plaque staining, red staining pads (Directa, Upplands Väsby, Sweden) using approved food coloring were used for all dogs.

Rewards given to the dogs when handled were liver slices (Dogman), lamb/chicken cubes (Four Friends), and soft cheese with a taste of smoked reindeer (Fjällbrynt). Rewards were given when the dog jumped up on the examination table, before, after one minute (at time for changing device) and after brushing. For the nonparticipant dogs similar amount of rewards was served in their pens.

## 4. Study Design

The dogs were divided by the group compositions they had in their existing pens. The treatment was performed in a separate examination room in their usual environment. The dental cleaning was performed by the same two persons (last/third-year veterinary nursing students) per study (toothbrush and textiles, respectively). Rewards were given to the dogs when handled. On each occasion, the dog’s identity was verified via microchip scanning or visual reading of earmarks.

## 5. Assessment of Oral Health

The assessment was performed before and after the treatment period by the author KBE. The buccal side of the maxillary canine tooth and the premolars were assessed before and after treatment, without sedation or anesthesia. The pre-treatment assessment was performed on the dentition as a whole since no dog showed any differences between the left and right side of the mouth at the start of the study. Post-treatment assessment was performed for each side of the mouth separately. The protocol used (Table 3) assessed gingival health, according to Löe 1967 [21], and plaque and calculus, according to Bellows 2019 [22]. Probing was not assessed.

The veterinarian was blinded to treatments and dog identity, and additionally assessed five nonparticipant dogs at the initial assessment and seven dogs at the final assessment. The nonparticipant dogs were not included in either study but to further blind the examiner. Three of the nonparticipant dogs had their teeth cleaned under anesthesia during the test period (5 days (nonparticipant 5), 26 days (nonparticipant 2) and 33 days (nonparticipant 1) prior to the final assessment). Photographs were taken of all dogs from the right and left side during the assessment before and after the treatment.

## 6. Staining

Plaque staining was performed according to the manufacturer’s instructions on all dogs before the experiment and on day 34 with red plaque staining pads. Photographs were taken of the dogs’ teeth from the right and left sides on both occasions to visualize the amount of plaque. For the statistics, the estimated PI was used.

## 7. Fear, Anxiety, and Stress (FAS) Scale

The FAS scale is compiled by Debbie and Kenneth Martin through Fear Free Pets (2018). The scale primarily describes early behavioral changes as well as physiological changes that the dog shows during stress and fear. One week before the start of the study, the dogs were trained daily to be handled on a table. The nonparticipant group was not handled but were assigned the same treats. The FAS stress level was assessed on the table during brushing on each occasion and a pre-set braking point if the dog reached a clear level 4.

The FAS scale categorizes fear, anxiety, and stress in three different levels, i.e., low (0–1), moderate (2–3), and high (4–5), and recommends that treatment should be discontinued. A dog that is judged to be 0 on the FAS scale shows no signs of stress or fear, has a relaxed body language, interacts with people, and willingly accepts treats. At level 1, 1–2 mild signs of stress or fear are seen, and they should not be seen more than three times per minute. Examples of these signs are licking around the mouth, avoiding eye contact, lifting the paw, turning the head away without moving the body, panting but commissures of lips are relaxed, or slightly dilated pupils. The dog should still be willing to accept treats and to interact with humans. At level 2, 1–2 signs of stress or fear are seen up to four times per minute. These signs may be that the ears are angled slightly backwards or to the side, the tail hangs down, the eyebrows are wrinkled, slow movements, overly attention seeking, and/or panting with a tighter mouth. The dog is still willing to accept treats and to interact with people. At level 3, more than two signs of stress or fear are seen, occurring more than four times per minute. The signs are the same as for level 2, but the dog may refrain from receiving treats or take it carefully. The dog may also be hesitant to interact with humans but not be actively evasive. Level 4 shows severe signs of stress or fear such as trying to escape, shaking, stiffening, dilated pupils, excessive panting, tense closed mouth, increased respiratory rate, ears angled backwards, and tail between legs. The dog may be reluctant to accept treats, not interested in interacting with humans, and may try to escape the situation but show no signs of aggression. At level 5, most signs of stress or fear are shown, including aggression, and when the dog is also unwilling to cooperate. Signs of aggression can be lunging, barking, growling, or biting.

FAS was assessed by the two veterinary nursing students performing the different treatments (tooth brushing and textiles) together during the tooth cleaning each day. Since two different methods were employed for each dog, FAS score changes could, by design, not be attributed to one of the two treatments, i.e., which one of the two toothbrushes (manual or ultrasonic) affected FAS.

## 8. Statistical Analysis

Total oral health was calculated by adding the gingiva index (GI), the plaque index (PI), and the calculus index (CI). GI, PI, CI, total oral health, and FAS values were analyzed by pairwise, a one-sided Wilcoxon signed rank test in R v 4.0.5 which compares before vs after treatment values. The limit of significance was set to *p* ≤ 0.05.

## 9. Results

### 9.1. Oral Health

All four methods of dental cleaning showed a statistically significant improvement in total oral health (*p* < 0.003). Separately analyzed, GI improved significantly for all four methods: manual toothbrush (*p* = 0.010, Figure 1), ultrasonic toothbrush (*p* = 0.018, Figure 1), nylon textile (*p* = 0.044, Figure 2), and microfiber (*p* = 0.018, Figure 2). PI also improved significantly for all groups (*p* < 0.002; Figure 1, Figure 2, Figure 3 and Figure 4). No significant improvement in CI was shown for any of the methods, i.e., the manual toothbrush (*p* = 0.087), the ultrasonic toothbrush (*p* = 0.074), the nylon textile (*p* = 0.5), or microfiber (*p* = 0.5)). However, for four individual dogs with grade-2 CI, the score had decreased to 1 and, in one case, to 0 after treatments (three dogs with toothbrush and one dog with textile). Examples of decrease in calculus amount are shown in Figure 5 and Figure 6. For individual scores, see Table 4.

### 9.2. Dental Disclosing Pads

When staining for dental plaque with dental disclosing pads before and after treatment, it was visible that the plaque had been reduced after using all methods (Figure 3 and Figure 4).

The photographs from the staining pre-and post-treatment were compared visually. All dogs treated with dental cleaning had visibly less plaque at the second staining (Figure 3 and Figure 4). Nonparticipant dogs had a comparable amount of plaque on both occasions.

### 9.3. Fear Anxiety and STRESS (FAS)

The dogs’ FAS levels were compared during dental cleaning day 1 and day 35. FAS decreased significantly (*p* < 0.011) for both the group of dogs (toothbrush and textile). The daily mean FAS divided in the groups are presented in Figure 7.

## 10. Discussion

### 10.1. Dental Health Assessment

A majority of dogs suffer from poor dental health [6,13]. Although knowledge may be increasing among dog owners regarding the importance of brushing their dog’s teeth, it can be difficult to initiate and maintain the habit of daily tooth brushing [14]. The present study shows that active dental home care during a relatively short period can improve dental health parameters, especially gingival health and the presence of dental plaque, and that dogs’ level of stress decrease during dental care training. The improvement in gingival health is in accordance with previous studies on tooth brushing, which have shown that gingivitis may be reversed in ten days with daily tooth brushing [23], but require continuous brushing for maintenance [24,25]. Dental plaque starts to form immediately after dental cleaning and dental calculus may start to form in 2–3 days without tooth brushing [6].

New methods are emerging on the market that are claimed to be as or more effective than manual cleaning with a toothbrush, but there is often a lack of scientific studies on the products. Animal health personnel shall perform their duties in accordance with science and proven experience. In practice, this means that products that do not have scientific support should not be recommended by veterinarians or veterinary nurses to animal owners. However, despite being widely used and recommended, there are no previous scientific publications on the use of textile for dental home care in dogs.

The amount of calculus decreased in some dogs for all methods, but the difference only showed significance for the ultrasonic toothbrush. In some of the dogs, flakes of calculus came off during the treatment period (Figure 1); however, the study period was not sufficiently long to determine whether any of the dental cleaning methods may eliminate calculus. In addition, all dogs had permanent access to rawhide bones which may help to decrease dental calculus [26,27,28].

Subjective comparisons were made from photos of all dogs’ teeth taken before and after five weeks of treatment, with and without plaque disclosure staining, as a qualitative confirmation of the dental health protocol. This visual assessment showed a clear decrease in plaque and gingivitis, and a decrease in calculus in several dogs (Figure 5 and Figure 6). This decrease in calculus was unfortunately not captured in the protocol.

### 10.2. Ultrasonic Toothbrush

This is, to the authors’ knowledge, the first scientific study where an ultrasonic toothbrush was used in dogs. The results showed a comparable result to that of the manual toothbrush, in agreement with previous studies on humans [16,17].

According to the manufacturer, the ultrasonic toothbrush can reach as far as 12 mm into the gums and into the smallest cavities around the teeth. The fact that ultrasound technology can reach hard-to-reach places that a manual toothbrush cannot handle could be an advantage in the case of established periodontitis with periodontal pockets. However, this was not investigated in the present study. According to international guidelines, professional dental cleanings and examinations under general anesthesia are recommended to address potential dental problems before initiating active dental home care [13,29]. According to the manufacturer, the ultrasonic toothbrush should also be suitable for dogs who experience difficulty with manual tooth brushing as it should only be kept still against the teeth. When the ultrasonic toothbrush is to be kept completely still, there may be difficulties if the dog does not want to sit still or feels uncomfortable having an object in his mouth and therefore chews during the treatment. This means that the suitability of the method varies depending on the dog’s individual characteristics. However, it is not clear if motion affects the ultrasonic cleaning or not.

### 10.3. Textiles

It has recently been shown that as many as 28% of Swedish dog owners sometimes use textiles to clean their dogs’ teeth, and that it is common that veterinarians and veterinary nurses to recommend this practice (12). The results from the present study indicate a comparable effect of the daily use of textiles to that of daily tooth brushing, which is the current gold standard for dental home care.

Textiles are generally assumed to not clean as well as a toothbrush between teeth or in the gum line; this could not be confirmed in the present study. However, in the photographs taken at the second staining, immediately after the dental cleaning, the plaque on and between the teeth was clearly reduced. How well the products can be accessed between the teeth is probably affected by how tight and oblique the teeth are and how carefully the cleaning is performed. Even so, textiles do not reach below the gum line and cannot clean periodontal pockets, and the recommendation may therefore be used with caution in patients with established periodontitis. More studies are needed to evaluate the use of textiles to prevent periodontal disease, but the present study shows promising results.

Both textile products were easy to use according to the performers’ subjective experiences and were generally well tolerated by the dogs, and the products were assessed as user-friendly. A high tolerance of the dogs facilitated the use of the products due to the cooperation of the dog. However, the products are clearly less suitable for dogs that have a tendency to bite.

When cleaning teeth, it is important to access the gum line as plaque in this area causes inflammation that can lead to periodontitis [6]. The manufacturer of the nylon glove advises against using it on the gingiva as the coarse structure may cause damage. It was perceived as very difficult to clean only the tooth without accessing the gingiva; however, no negative effect of this, such as redness or swelling, was assessed after 35 days of daily use. How hard the glove is pressed against the teeth and gingiva and how long it is used every day may have a bearing on whether injuries can occur.

The microfiber finger cloth was perceived as flexible and took up little space in the dogs’ mouth, and it could be put on any finger. The microfiber material was soft and was sometimes difficult to feel the demarcation between tooth and gingiva and whether the lip or tongue came in between. There was a risk that the finger cloth could be swallowed by the dog if dropped. The manufacturer recommends it to be used as an introduction to tooth brushing with a toothbrush, after surgery in the mouth when a softer material is beneficial or to a dog who does not like the toothbrush.

### 10.4. FAS and Dental Home Care

None of the dogs in this study previously had their teeth brushed on more than single occasions. All dogs included are used in teaching for veterinary and veterinary nursing students and have adapted to the handling by different people but in different degrees, as some were born at the university and some were imported. The veterinary nurse students who performed the dental cleaning were also those who assessed the FAS level in the dogs. It can be questioned whether the assessments were completely objective or whether adjustments were made during the course of the study.

There can be several reasons why pet owners have difficulty starting to brush their dog’s teeth, including that the pet owner experiences that the dog becomes stressed or that there is a conflict situation between the dog and the owner. It can sometimes be difficult to perceive the signals the dog shows when it becomes stressed or scared [30,31,32], especially if they are subtle signals.

In this study, the effect of the electric toothbrush could not be studied as the dogs showed FAS reactions above the predetermined breaking point already at the initial habituation of the dogs. Manual tooth brushing with an electric toothbrush turned off was therefore the method used in the study. The consequence was that the study could not study the effect of electric toothbrush on oral health. It was also not possible to draw any conclusions about whether the dogs would have become accustomed to the electric toothbrush during a longer period of training by desensitization [33]. Electric toothbrushes have been shown to have a better effect than manual toothbrushes in both the long and short term, in terms of plaque and gingivitis [34,35].

The FAS level for each dog at the start of the treatment depends mainly on how accustomed they were to this type of handling. Some of the dogs included in the experiment were older and have been involved in teaching at the university for many years. They are adapted to meet new people and being handled and were given a low FAS throughout the study. Ten dogs were younger and were not as used to handling, but most of them were sociable and forward. Two of them were very shy and had therefore not been included in teaching for students but they made great progress in the daily handling during the present study; however, they maintained a high FAS score during treatment on the examination table. It is assumed that individual differences between dogs affect habituation time [36].

In the study, there was a significant difference in the dogs’ FAS values compared between the first and last treatment occasion after five weeks of daily treatment for all four treatments. The study thus showed that habituation took place for the methods used. This information can be useful for professional veterinary nurses when they want to motivate dog owners to start brushing their dog’s teeth.

Some dogs showed several stress-related behaviors during the treatment with the ultrasonic brush, which did not seem as pronounced during the manual tooth brushing. The behaviors noted were attempts to get away with the mouth, chewing on the toothbrush, and beating away the toothbrush with the paws. This could be due to the fact that the ultrasonic toothbrush must be kept completely still, and some dogs have more difficulty sitting still for that period of time. The included toothpaste for the ultrasonic toothbrush was perceived by the study leaders to have a strong scent and the dogs visibly reacted to the toothpaste at the beginning of the study by sniffing, smacking and wrinkling their lips. The fact that the toothpaste seemed to affect the dogs could also be a reason why some dogs were perceived to show a higher FAS level around the ultrasonic toothbrush initially. In addition, for all dogs, the surrounding environmental factors, such as dog barking, could increase the FAS score occasionally.

Furthermore, according to the manufacturer, the ultrasonic toothbrush is completely silent and vibration-free, but it has not been possible in the study to clarify whether the ultrasonic toothbrush emits high-frequency sounds or vibrations that are only heard by the dogs. This could be another possible cause of the more frequent aversive behaviors. According to the manufacturer’s instructions, it is sufficient to use the ultrasonic toothbrush on the teeth twice a week to maintain oral health. However, the recommendation from human dentistry is to use it as often as a manual toothbrush, i.e., daily, which should be noted. With a large amount of calculus on the teeth, the ultrasonic toothbrush may be held for a longer time against each tooth to facilitate calculus breakdown, according to the manufacturer. However, this was not tested in this study. Furthermore, whether a prolonged contact can affect the teeth or gingiva is not known.

A flavored dog toothpaste may facilitate brushing or cleaning with textiles by giving the dog a positive experience when cleaning teeth. It can also complicate the treatment because the dog likes to lick and chew while cleaning. In the present study, toothpaste was not used with textiles to evaluate purely the mechanical effect. The difficulties that the study showed with using an electric toothbrush on dogs can be a reason why pet owners do not choose to brush their dog’s teeth. The dog may, as the study showed, experience it as stressful and, for the user, it then becomes very difficult to be able to brush their dog’s teeth properly with a switched-on electric toothbrush. A manual toothbrush may be easier than an electric toothbrush to get started with continuous toot brushing.

When cleaning teeth with textiles, some dogs seemed to react more negatively by the glove than by the finger cloth through avoidant behaviors such as pulling the head away. A possible explanation for this may be that the material in the glove is coarser and the finger cloth is softer.

### 10.5. Strengths and Limitations

The parameters assessed in the dental protocol were in four levels (0–3), which made it difficult to assess small differences that occurred after five weeks of tooth brushing. It would have been desirable to have a more detailed assessment scale with more levels. The assessments were performed on fully conscious dogs so, if not sedated or anesthetized, it is not possible to perform a complete examination even though the protocol also could be more sensitive to catch small differences. Some of the dogs received the same degree of calculus or gingivitis during the assessments, despite differences in the number of affected teeth. As an example, a dog with calculus on one tooth received the same assessment in the protocol as a dog with calculus on several teeth. Dogs with calculus directly adjacent to the gums were assigned the same level as dogs where the calculus only sat further down on the tooth. This has a clinical difference in that calculus that is not in contact with the gums has a lower impact on gingival health. Nonparticipant dogs were assessed as having plaque and gingivitis, as opposed to dogs in the treatment groups. Because the treatment was blinded to the assessing veterinarian, validity of the used protocol could be ensured.

In the study, only beagle dogs were used, as other breeds may differ in FAS and in dental health. The result should be applicable to other dog breeds with similar size and head shape. In a smaller or brachycephalic breed, it may be more difficult to access the products used in the study because of malocclusion or crowded teeth. In addition, privately owned dogs with a stronger bond to their owner may differ in training/habituation of tooth brushing [37].

The principle of the ultrasonic toothbrush is that ultrasonic waves replace the need for mechanical motion and that the toothbrush should only be kept still against the tooth. During the study, the dogs did not always sit still with their head/mouths, but chewed and licked, which meant that the toothbrush moved to the teeth unintentionally to varying degrees. In this way, a mechanical cleaning of the teeth has occurred on several occasions. We cannot exclude that the positive results also depend on the mechanical motion and not only the ultrasonic technology.

## 11. Conclusions

All four tested treatments for dental home cleaning (manual and ultrasonic toothbrush and nylon and microfiber textile) reduced the amount of plaque and gingivitis after five weeks of daily use. A reduction in calculus was seen in four dogs.

This knowledge may be useful for veterinary staff in contact with pet owners to demonstrate the importance of brushing their dog’s teeth frequently at home especially after a professional dental cleaning. As the study also showed that there was a habituation over time in the dogs regarding the treatments, this could be used by the veterinary staff in communication with dog owners to inspire and motivate them to initiate and continue to brush their dog’s teeth.

## Figures and Tables

**Figure 1 animals-11-02481-f001:**
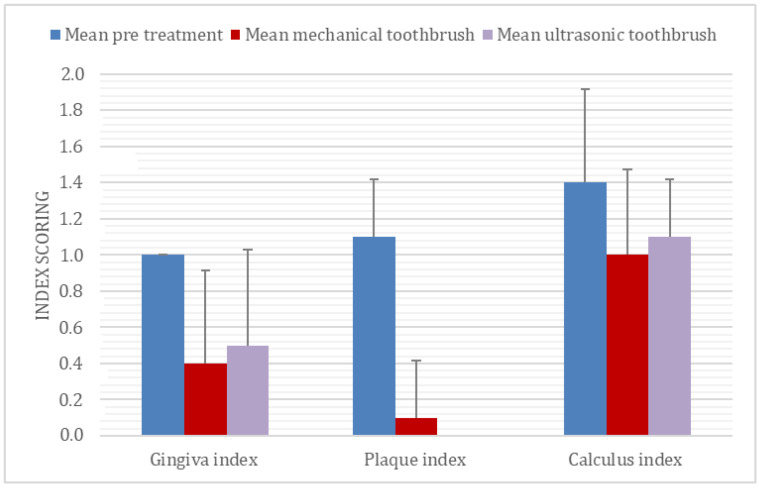
Mean ± SD before and after 35 days of daily dental cleaning with manual or ultrasonic toothbrush in 10 beagle dogs.

**Figure 2 animals-11-02481-f002:**
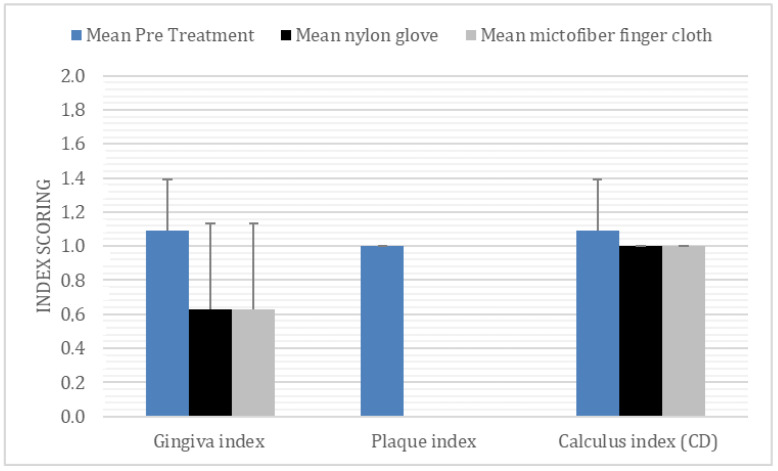
Mean ± SD before and after 35 days of daily dental cleaning with nylon glove or microfiber finger cloth n 11 beagle dogs.

**Figure 3 animals-11-02481-f003:**
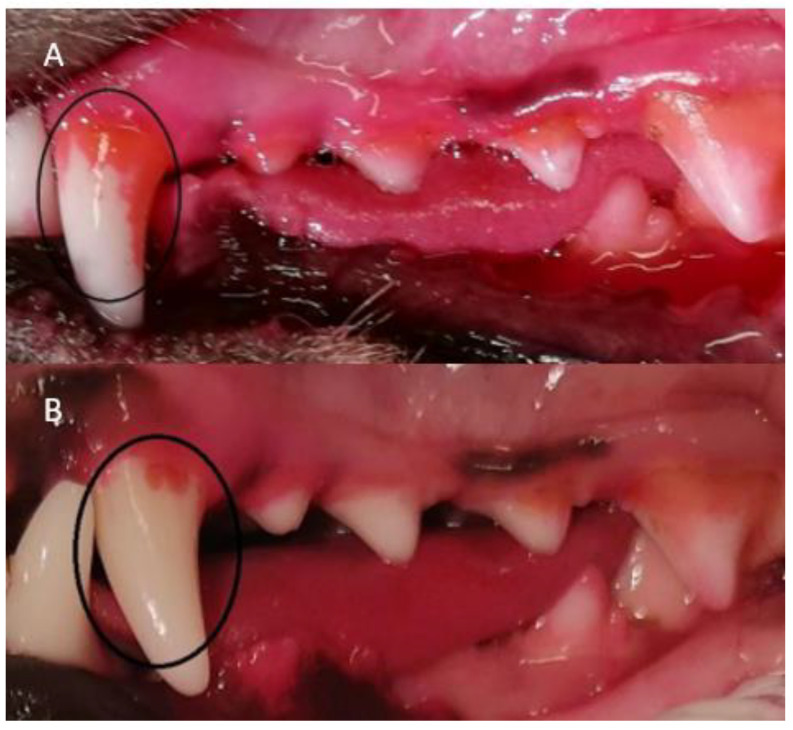
(**A**): Staining before treatment in dog ‘Toothbrush 6′. Red colour depicts plaque. (Photo: Fanny Eriksson) (**B**): Staining of the left side of dog ‘Toothbrush 6′ after dental cleaning with an ultrasonic toothbrush. The amount of plaque has decreased on tooth 204 at the final staining. Small remnants of plaque can still be seen on some teeth. (Photo: Sara Wiman).

**Figure 4 animals-11-02481-f004:**
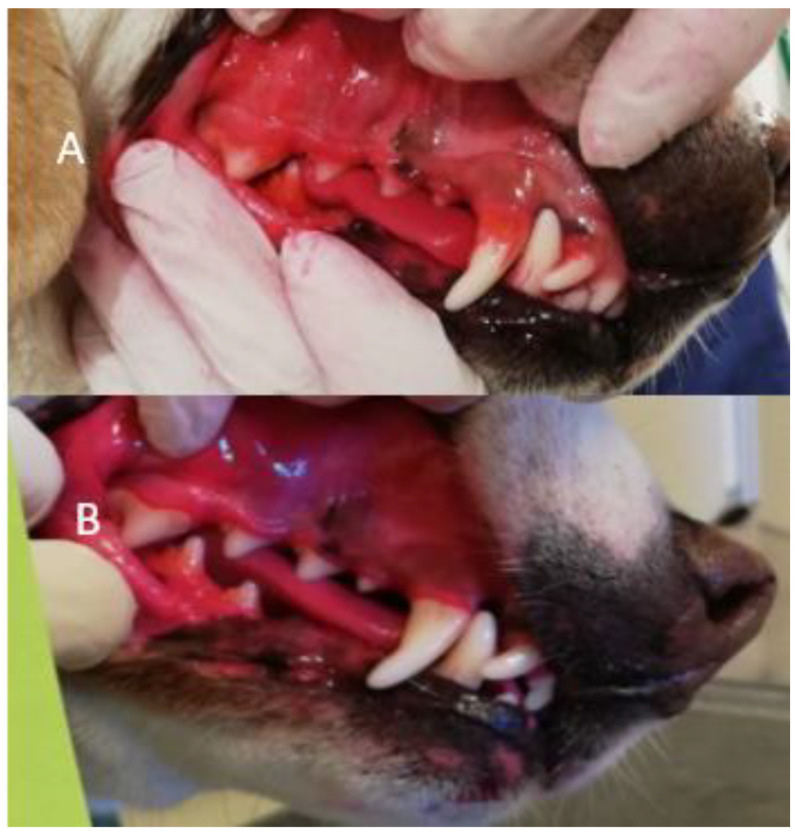
Dog ‘Textile 1′ (**A**): first staining; and (**B**): after 35 days with daily dental cleaning with microfiber finger cloth. The bacterial plaque is stained red. (Photo: Fanny Eriksson).

**Figure 5 animals-11-02481-f005:**
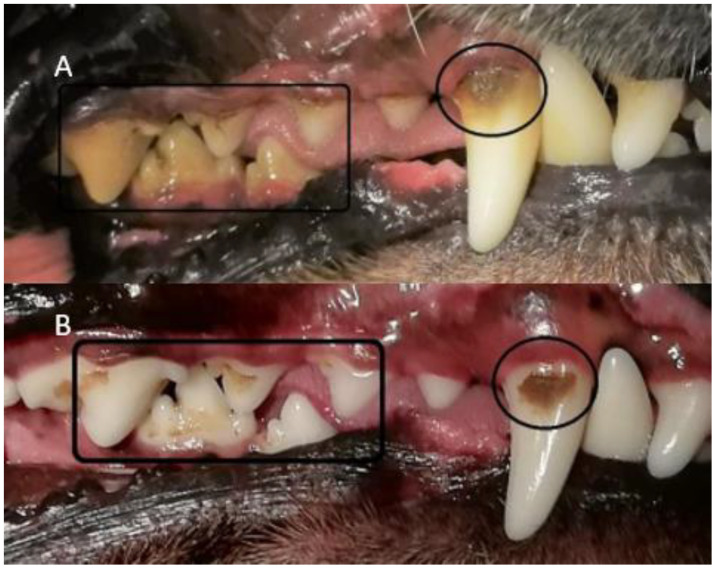
(**A**) The amount of calculus including plaque on the right side at the first assessment before dental cleaning has begun on Toothbrush 3. (Photo: Fanny Eriksson) (**B**) The amount of calculus on the right side at the second assessment on Toothbrush 3 after dental cleaning with manual toothbrushing. The amount of calculus has visibly decreased on all teeth. (Photo: Fanny Eriksson).

**Figure 6 animals-11-02481-f006:**
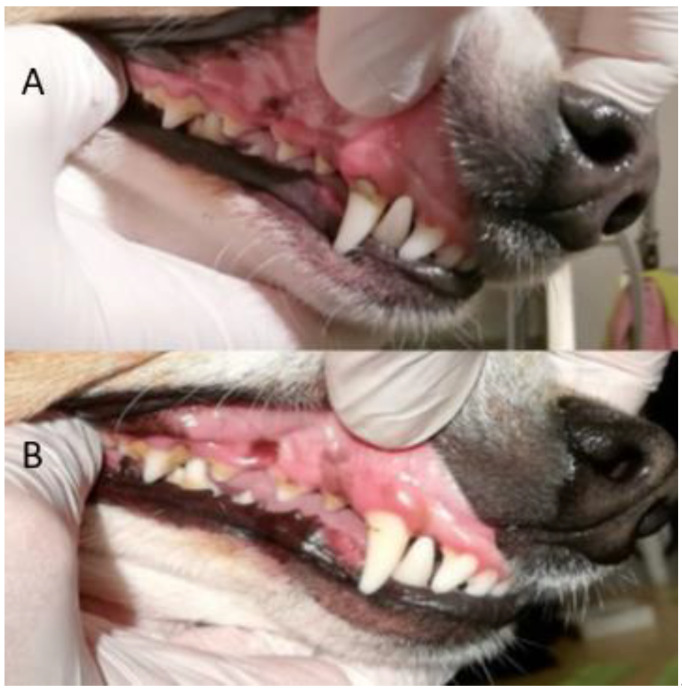
For dog ‘Textile 4′ the amount of calculus decreased during the study. (**A**): The teeth at the first assessment before the start of dental cleaning. Calculus can be seen on several teeth. (**B**): In this last assessment, calculus decreased on tooth 104 and, to some extent, on tooth 108 compared to (**A**) (Photo: Fanny Eriksson).

**Figure 7 animals-11-02481-f007:**
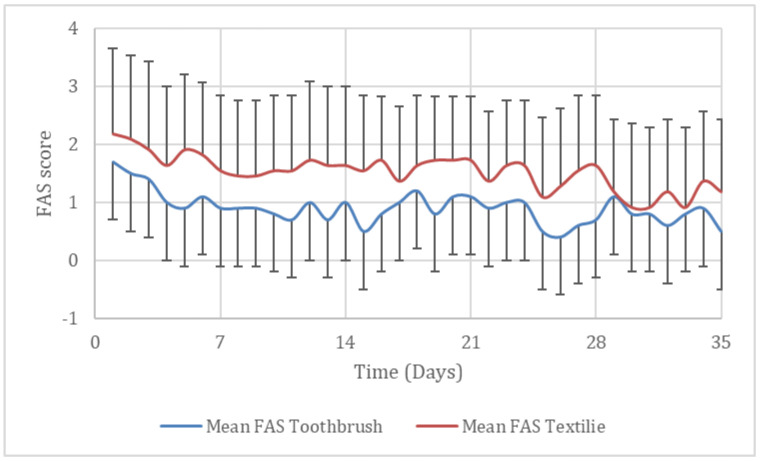
Mean FAS score ±SD for beagle dogs during dental cleaning with toothbrush (mechanical and ultrasonic, N = 10) or textiles (nylon glove and microfiber finger cloth, N = 11) during 35 consecutive days.

**Table 1 animals-11-02481-t001:** Age, sex, and treatment allocation of participating dogs.

Dog ID	Age (Years)	Sex	Treatment Right Side	Treatment Left Side
Toothbrush 1	3	Female	Manual	Ultrasonic
Toothbrush 2	3	Female	Manual	Ultrasonic
Toothbrush 3	3	Female	Manual	Ultrasonic
Toothbrush 4	3	Female	Manual	Ultrasonic
Toothbrush 5	3	Female	Manual	Ultrasonic
Toothbrush 6	2	Female	Ultrasonic	Manual
Toothbrush 7	2	Female	Ultrasonic	Manual
Toothbrush 8	4	Female	Ultrasonic	Manual
Toothbrush 9	3	Female	Ultrasonic	Manual
Toothbrush 10	12	Female	Ultrasonic	Manual
Textile 1	7	Male	Microfiber	Nylon
Textile 2	2	Male	Microfiber	Nylon
Textile 3	2	Male	Microfiber	Nylon
Textile 4	2	Female	Nylon	Microfiber
Textile 5	2	Female	Microfiber	Nylon
Textile 6	2	Female	Microfiber	Nylon
Textile 7	2	Female	Nylon	Microfiber
Textile 8	11	Female	Nylon	Microfiber
Textile 9	10	Female	Nylon	Microfiber
Textile 10	2	Female	Nylon	Microfiber
Textile11	2	Female	Nylon	Microfiber

**Table 2 animals-11-02481-t002:** Additional dogs used to blind evaluation.

Dog ID	Age (Years)	Sex	Professional Dental Cleaning	Evaluation
Nonparticipant 1	9	Male	Day 2	Day 0 and 35
Nonparticipant 2	7	Male	Day 9	Day 0 and 35
Nonparticipant 3	9	Female	Day 30	Day 0 and 35
Nonparticipant 4	4	Male	None	Day 0 and 35
Nonparticipant 5	4	Male	None	Day 0 and 35
Nonparticipant 6	9	Female	None	Day 35
Nonparticipant 7	9	Female	None	Day 35

**Table 3 animals-11-02481-t003:** Oral health assessment protocol: gingival health index (GI), plaque index (PI), and calculus index (CI).

**Gingival Health Index (GI)**	
0	no inflammation
1	mild inflammation, mild hyperemia
2	moderate inflammation, moderate hyperemia
3	severe inflammation, severe hyperemia, swelling, bleeding spontaneously, ulcerations
**Plaque Index (PI)**	
0	no plaque
1	thin layer of plaque along the gingival edge
2	moderate layer of plaque and/or plaque in sulcus
3	abundant plaque and soft material in sulcus
**Calculus Index (CI)**	
0	no calculus
1	supragingival or calculus that extends only slightly below the free gingival margin
2	moderate amount of supra- and/or subgingival calculus or only subgingival calculus
3	abundant supragingival and/or subgingival calculus

**Table 4 animals-11-02481-t004:** Individual scores for gingiva index (GI), plaque index (PI) and calculus index (CI) and FAS score at the beginning and end of 35 days of daily dental cleaning (no cleaning for nonparticipant dogs).

Dog ID	Before Treatment (GI, PI, CI)	Ultrasonic Toothbrush (GI, PI, CI)	Manual Toothbrush (GI, PI, CI)	FAS before Treatment Period	FAS at End of Treatment Period
Toothbrush 1	1, 1, 1	0, 0, 1	0, 0, 1	1	1
Toothbrush 2	1, 1, 2	1, 0, 2	1, 0, 2	1	2
Toothbrush 3	1, 1, 2	1, 0, 1	0, 0, 0	1	0
Toothbrush 4	1, 1, 1	0, 0, 1	0, 1, 1	2	0
Toothbrush 5	1, 1, 1	0, 0, 1	0, 0, 1	2	1
Toothbrush 6	1, 1, 1	1, 0, 1	1, 0, 1	3	0
Toothbrush 7	1, 1, 2	1, 0, 1	1, 0, 1	3	1
Toothbrush 8	1, 1, 1	1, 0, 1	1, 0, 1	1	0
Toothbrush 9	1, 1, 1	0, 0, 1	0, 0, 1	3	0
Toothbrush 10	1, 2, 2	0, 0, 1	0, 0, 1	1	0
		Nylon (GI, PI, CI)	Microfiber (GI, PI, CI)		
Textile 1	1, 1, 1	1, 0, 1	0, 0, 1	0	0
Textile 2	1, 1, 1	1, 0, 1	0, 0, 1	3	1
Textile 3	1, 1, 1	1, 0, 1	1, 0, 1	1	1
Textile 4	1, 1, 1	1, 0, 1	1, 0, 1	3	1
Textile 5	2, 1, 2	0, 0, 1	1, 0, 1	3	1
Textile 6	1, 1, 1	1, 0, 1	1, 0, 1	3	1
Textile 7	1, 1, 1	0, 0, 1	0, 0, 1	2	1
Textile 8	1, 1, 1	1, 0, 1	1, 0, 1	0	0
Textile 9	1, 1, 1	1, 0, 1	0, 0, 1	1	0
Textile 10	1, 1, 1	0, 0, 1	1, 0, 1	4	4
Textile 11	1, 1, 1	0, 0, 1	1, 0, 1	4	3
		No treatment (GI, PI, CI)			
Nonparticipant 1	1, 1, 1	1, 1, 1	-	NA	NA
Nonparticipant 2	1, 1, 2	1, 1, 0	-	NA	NA
Nonparticipant 3	1, 1, 2	1, 1, 1	-	NA	NA
Nonparticipant 4	1, 1, 1	1, 1, 1	-	NA	NA
Nonparticipant 5	1, 1, 1	0, 0, 0	-	NA	NA
Nonparticipant 6	NA	0, 0, 1	-	NA	NA
Nonparticipant 7	NA	1, 0, 1	-	NA	NA

## Data Availability

Not applicable.

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
