# Peer review of "Improved Oral Health and Adaptation to Treatment in Dogs Using Manual or Ultrasonic Toothbrush or Textile of Nylon or Microfiber for Active Dental Home Care"

_animals, 2021, doi:10.3390/ani11092481_

Round 1

Reviewer 1 Report

Dear authors

Here my contribution to clarify some aspects of the work.

Simple Summary 

Veterinary dentistry language is very specific and that's why I suggest some corrections to respect the essence of the evoked concepts. Please change the summary according to these suggestions.

I prefer to use the term "oral disease" rather than "dental disease" because as our focus is periodontal disease the tissues affected are periodontal rather than dental.

Page 1, Lines 9-10 - Poor oral health is a common problem that course with pain and harms the wellbeing of affected dogs.

The method is tooth scaling and toothbrush is a device for this method performing.

Page 1, Lines 10-12 - In this study, four different devices used for active  home dental care were evaluated; two toothbrushes, manual and ultrasonic, and two textile products, a nylon glove and a microfiber finger cloth.

Page 1, Lines 12-13 - The equipments were evaluated for efficacy against  plaque and calculus, using twenty one beagle dogs.

Page 1, Lines 16-17 - All devices have been shown to improve the oral health in the dogs, reducing plaque and gingivitis as a result. Furthermore, the amount of calculus has also decreased in some individuals.

Introduction

Page 2, Lines 48, 50 - It would be very important to define which tissues are affected during both phases of periodontal disease evolution.

Materials and Methods

It is important to well define the study groups.

The purpose of using control animals in this context does not seem clear or well explained in the text. Some animals were subject to scaling during the course of the work and on the others there is no reference!

Page 2, Line 90 - Population (teaching dogs at the Swedish University of Agricultural Sciences (SLU) in Uppsala) and sample studied instead of dogs.

Page 3, Lines 98, 99 - One of the foods is recommended for dogs over 7 years old, it should explain who ate what.

Page 3, Line 106 - Textile 4 to Textile 11 Female in italic?

Materials

Page 4, Line 108 - 

The authors intend to compare the use of different equipment to clean the oral cavity. However, how can one compare, for example, the difference between the use of a manual toothbrush and an ultrasonic toothbrush, if there are other different variables, such as toothpaste?

Page 5, Line 160 - Probing was not assessed.

Page 5, Line 161- Gingival Index and Gingival Index (GI) in the table

Page 6, Line 202 - Signs of aggression can be lunging, barking, growling, biting or biting.

Is erythrosine the dye used to reveal plaque ?

Results

Figures 1 and 2 were not previously mentioned in the text.

Page 10, Line 246 - Figure 4 - "The bacterial plaque was stained red" instead "The red is plaque"

I still don't understand the advantage of using a control group that was subjected to scaling.

Conclusions

I believe the conclusions are very vague and do not reflect the results of the work. Are all equipment equally valid? For each context which do you prefer? Which causes the most stress in the animal?

There are many issues not yet fully explored in this work that has potential.

Author Response

Dear Sir/Madam,

Enclosed please find our manuscript Manuscript ID: animals-1297335, title: Improved oral health and adaptation to treatment in dogs using manual or ultrasonic toothbrush or textile of nylon or microfiber for active dental home care.

Content of the submission:

  1. Response to the reviewers
  2. Revised Manuscript

We thank you for your assistance with this manuscript.  If any problems still exist, please contact me at once and I will do everything I can to rectify the problem. 

The response to authors is as follows:

Simple Summary 

Veterinary dentistry language is very specific and that's why I suggest some corrections to respect the essence of the evoked concepts. Please change the summary according to these suggestions.

I prefer to use the term "oral disease" rather than "dental disease" because as our focus is periodontal disease the tissues affected are periodontal rather than dental.

Point well made, oral will be substituted for where appropriate

Page 1, Lines 9-10 - Poor oral health is a common problem that course with pain and harms the wellbeing of affected dogs.

accepted

The method is tooth scaling and the toothbrush is a device for this method performing.

That is correct, but, referring to the question under Material, further down, we will discuss the choice of terminology:

The authors intend to compare the use of different equipment to clean the oral cavity. However, how can one compare, for example, the difference between the use of a manual toothbrush and an ultrasonic toothbrush, if there are other different variables, such as toothpaste?

The toothpaste sold in conjunction with the Emmipet Ultrasonic is specified for the purpose, as the cleaning action is the combined effect of the device and the toothpaste: “ form[ing] millions of microbubbles in the mouth in every fraction of a second. The microbubbles penetrate into the interdental spaces as well as into the smallest gaps and implode”

 If a dog owner chooses this method, they will be using the two together. Thus, comparing the use of an ultrasonic brush with anything else would need to include the dentifrice.

If an alternative, manual toothbrush is chosen, the dog owner is not likely to opt for the more expensive toothpaste made specifically for a different, ultrasonic product.  Petosan is a widely used and easily accessible toothpaste, recommended by veterinarians for use in combination with a toothbrush. Thus the combination of the manual toothbrush and Petosan is the comparable substitution.

This may in part be the explanation why we had elected to refer to the two different approaches as “methods” rather than “devices” or “equipment” since we refer to the whole sets: brush plus dentifrice plus either of the two different means of descaling by either manually massaging with the bristles or holding the ultrasonic brush in one position at a time (once daily during 35 days).

Thus, we are not per se comparing the brush vs the ultrasonic device, but the effects of the selected descaling method. The dogs are used as their own controls (before and after treatments) and no treatment are compared to the other.  For the dog owner, the important information is whether the ultrasonic brush / manual brush/textile used as per the instructions, result in better oral health, and how it compares to no descaling. Hence, the individual effects of the dentifrice vs the bristles vs the manual manipulation, while important, could not be incorporated in a study this small, but can be covered in additional studies. Text to this effect will be added to the discussion.

We agree however that substituting “device” or “equipment” for “method” makes sense, and have adjusted the text accordingly.

Page 1, Lines 10-12 - In this study, four different devices used for active  home dental care were evaluated; two toothbrushes, manual and ultrasonic, and two textile products, a nylon glove and a microfiber finger cloth.

accepted:
methods -> devices.
Not changed from active home dental care from active dental home care.

Page 1, Lines 12-13 - The equipments were evaluated for efficacy against  plaque and calculus, using twenty one beagle dogs.

adjusted: methods. Not adjusted: Gingivitis:

These pieces of equipment were evaluated for efficacy against gingivitis, plaque, and tartar, using twenty-one beagle dogs.

Page 1, Lines 16-17 - All devices have been shown to improve the oral health in the dogs, reducing plaque and gingivitis as a result. Furthermore, the amount of calculus has also decreased in some individuals.

Adjusted: Devices-
Not altered: “have been shown” - the tense suggests that this is a known fact from previous studies rather than the findings of this study. The word limit of 200 words, 202 in the original text. The additional “as a result” would be clarifying, had the word count allowed it.

Introduction

Page 2, Lines 48, 50 - It would be very important to define which tissues are affected during both phases of periodontal disease evolution.

Added: During early stage of gingivitis only the soft tissue is affected, whereas when it progresses to periodontitis also the bone structures supporting the teeth will be compromised.

Materials and Methods It is important to well define the study groups.

The purpose of using control animals in this context does not seem clear or well explained in the text. Some animals were subject to scaling during the course of the work and on the others there is no reference!

The additional dogs were not enrolled specifically for the purpose of serving as controls, with the same evaluations performed as the participants.
Rather, they were evaluated by a veterinarian not informed of which dogs were non participants, vs participated in the nylon / microfiber study. This is now clarified in the manuscript

Page 2, Line 90 - Population (teaching dogs at the Swedish University of Agricultural Sciences (SLU) in Uppsala) and sample studied instead of dogs.

Changed dogs to study population

Page 3, Lines 98, 99 - One of the foods is recommended for dogs over 7 years old, it should explain who ate what.

“The feed given was Hills Vet Essentials Adult Medium orHills Science Plan Mature Adult 7+ with chicken, according to age.”

Page 3, Line 106 - Textile 4 to Textile 11 Female in italic?

Changed

Materials

Page 4, Line 108 - 

The authors intend to compare the use of different equipment to clean the oral cavity. However, how can one compare, for example, the difference between the use of a manual toothbrush and an ultrasonic toothbrush, if there are other different variables, such as toothpaste?

See discussion above

Page 5, Line 160 - Probing was not assessed.

corrected

Page 5, Line 161- Gingival Index and Gingival Index (GI) in the table

corrected

Page 6, Line 202 - Signs of aggression can be lunging, barking, growling, biting or biting.

corrected

Is erythrosine the dye used to reveal plaque ?

The staining pads consists of approved food coloring

Results

Figures 1 and 2 were not previously mentioned in the text.

Added in the text

Page 10, Line 246 - Figure 4 - "The bacterial plaque was stained red" instead "The red is plaque"

corrected

I still don't understand the advantage of using a control group that was subjected to scaling.

See clarification above

Conclusions

I believe the conclusions are very vague and do not reflect the results of the work. Are all equipment equally valid? For each context which do you prefer? Which causes the most stress in the animal?

We did not compare the different treatments to each other only comparisons before vs after treatment was performed using the dogs as their own control so conclusions was drawn comparing the different equipment. The manufacturer of the textiles recommend their products for dogs not yet used to tooth brushing. As the FAS was estimated for the total procedure (tooth brushing or textiles) and the dogs was uneven in FAS initially we could not conclude which was causing most stress in the dogs. The aim was to investigate the habituation over time.

Reviewer 2 Report

This is an interesting study examining the efficacy of various strategies (toothbrush, ultrasonic toothbrush, two types of textiles) for daily cleaning of dog teeth.  All approaches were found to improve dental health over a 35 day period.  The authors also examined dog stress levels during ‘brushing’ with FAS before and at the end of the treatment period, and found that dog stress levels declined during the course of the treatment. Overall the paper is important and well written.  I have some concerns about the statistical approach, as detailed below, in addition to some minor recommendations to improve clarity.

L9 - change to 'can cause'

L15:  Change 'were' to 'was'

L53:  Change 'do' to 'does'

L88:  Change to '...whether dogs habituate to the procedure...'

L117 & 122:  Move info about donations to the acknowledgments section.

L119 & 123:  Was toothpaste used with either of the textile options?

L123:  Underline Microfiber to match other methods.

L126:  Change 'was' to 'were'

L126:  Can you please provide further information on the method used to assess plaque staining?

L134:  I'm assuming brushing occurred for one minute total including all areas?  Please clarify.

L148:  Can you provide some rationale for why toothpaste was not used with the textiles?  Also please include this information with the actual methods above as suggested earlier (so that it matches with the other methods).

L149:  How were rewards delivered during the actual tooth brushing procedure?  Immediately before and after?  More information would be helpful.

L166: change 'was' to 'were'

L170:  Suggest moving the information on plaque staining materials from above to this section - easier to follow and understand if everything is together.

L179:  When were the treats for the control group given?

L177-180: Were all of the dogs used in the pretraining period for this assessment, and does this mean that some or all of the dogs in the dental treatment groups received some oral care prior to starting the main study?  Or were dogs only handled during this period?  More information is needed on the exact training protocol (handling and treat use) with the dogs during the pre-treatment and treatment periods. 

Also, when were data collected for FAS analysis in the main study?  The text says that dogs were assess each day for this week, but the statistical analysis section indicates that dogs were assessed before and after the study period.  Please clarify.

L184:  Change 'sweets' to 'treats'.  Here and below.

L187-188:  What is mean by 'hoarseness with relaxed mouth pores'?

L181-202:  Who completed the FAS, and was it completed in real-time or from video?  Were the assessors blind to the study purposes or the treatment groups?

L203-208:  The data that were collected are categorical rather than continuous, so parametric t-tests aren't an appropriate approach for analysis.  Some might argue that parametric stats are ok for the total oral health score, although technically non-parametric approaches are still better since one can't assume that the distance between levels is consistent.  For the other outcomes with only 3-4 levels that are, again, not necessarily consistent between levels, parametric approaches are not a good fit.  The authors are encouraged to explore non-parametric approaches that are better suited to the data collected, and potentially consult with a statistician. Based on the raw data provided it is likely that the overall results will be similar with a non-parametric approach.

L211:  Were differences also examined for the control group as scores appear to be numerically lower for the second assessment?

Figures 1 & 2:  Replace commas with periods.  e.g., 0.2 rather than 0,2

L242-244: In order to discuss comparisons the amount of staining needs to be quantified in some way (scoring system or % tooth covered as measured with Image j or other similar software).  These pictures are striking but are not enough to draw conclusions from on the whole. Please provide specific details for the method of assessment and comparison.

L292: Details about rawhide bones access should be included in the Materials and Methods. Did access differ before and after treatment or was it consistently provided both before and after?  If only after it is a serious confound for the study, but if it was also provided before it is unlikely to explain the improvement that occurred.  Please update this section to clarify.

L327-328:  Based on the data provided no comparisons were made between the textile and toothbrush groups, so you can't really say that there were comparable effects - at most you can say that both groups showed improvements.  However, given that the studies were conducted at the same time you could consider statistically comparing the treatments between groups.

L339:  You did not present any data on the user experience therefore it isn't appropriate to include comments about this in the discussion.  Instead maybe you could refer to the average dog response which shows that the dogs tolerated the process without biting?

L358:  Previous studies have looked at habituating animals (including dogs) to different types of handling and procedures, and it would be helpful to incorporate some of this information into this portion of the discussion. Eg, https://doi.org/10.1016/j.vascn.2014.12.007, https://doi.org/10.3390/ani9100767,

L364-365:  Was reliability for scoring FAS assessed before or during the study?

L368-370:  A number of studies have assessed people's ability to detect these subtle signals - please add citations to support this statement. Eg, http://dx.doi.org/10.1080/08927936.2016.1228750, https://doi.org/10.1371/journal.pone.0051775

L394-396:  Again, this information is not based on data presented in the paper.  If this information was recorded please include it in the results section.

Author Response

Dear Sir/Madam,

Enclosed please find our manuscript Manuscript ID: animals-1297335, title: Improved oral health and adaptation to treatment in dogs using manual or ultrasonic toothbrush or textile of nylon or microfiber for active dental home care.

Content of the submission:

  1. Response to the reviewers
  2. Revised Manuscript

We thank you for your assistance with this manuscript.  If any problems still exist, please contact me at once and I will do everything I can to rectify the problem. 

The response to authors is as follows:

Reviewer 2

Comments and Suggestions for Authors

This is an interesting study examining the efficacy of various strategies (toothbrush, ultrasonic toothbrush, two types of textiles) for daily cleaning of dog teeth.  All approaches were found to improve dental health over a 35 day period.  The authors also examined dog stress levels during ‘brushing’ with FAS before and at the end of the treatment period, and found that dog stress levels declined during the course of the treatment. Overall the paper is important and well written.  I have some concerns about the statistical approach, as detailed below, in addition to some minor recommendations to improve clarity.

L9 - change to 'can cause'

Changed

L15:  Change 'were' to 'was'

Changed

L53:  Change 'do' to 'does'

Changed

L88:  Change to '...whether dogs habituate to the procedure...' 

Changed

L117 & 122:  Move info about donations to the acknowledgments section.

Changed

L119 & 123:  Was toothpaste used with either of the textile options?

No. Clarified in the text 

L123:  Underline Microfiber to match other methods.

Changed

L126:  Change 'was' to 'were'

Changed

L126:  Can you please provide further information on the method used to assess plaque staining?

More information and clarification added in the text

L134:  I'm assuming brushing occurred for one minute total including all areas?  Please clarify.

Yes 1 minute for each treatment so total 2 min per day per dog - clarified

L148:  Can you provide some rationale for why toothpaste was not used with the textiles?  Also please include this information with the actual methods above as suggested earlier (so that it matches with the other methods).

The toothpaste sold in conjunction with the Emmipet Ultrasonic is specified for the purpose, as the cleaning action is the combined effect of the device and the toothpaste: “ form[ing] millions of microbubbles in the mouth in every fraction of a second. The microbubbles penetrate into the interdental spaces as well as into the smallest gaps and implode”

 If a dog owner chooses this method, they will be using the two together. Thus, comparing the use of an ultrasonic brush with anything else would need to include the dentifrice.

If an alternative, manual toothbrush is chosen, the dog owner is not likely to opt for the more expensive toothpaste made specifically for a different, ultrasonic product.  Petosan is a widely used and easily accessible toothpaste, recommended by veterinarians for use in combination with a toothbrush. Thus the combination of the manual toothbrush and Petosan is the comparable substitution.

The material and method section is now updated and clarified

L149:  How were rewards delivered during the actual tooth brushing procedure?  Immediately before and after?  More information would be helpful. L179:  When were the treats for the control group given?

Rewards were given when the dog jumped up on the examination table, before, after one minute (at time for changing device) and after brushing. For the control dogs similar amount of rewards was served in their pens.

Information is added in the text

L166: change 'was' to 'were'

Changed

L170:  Suggest moving the information on plaque staining materials from above to this section - easier to follow and understand if everything is together.

Added information in the text

L177-180: Were all of the dogs used in the pretraining period for this assessment, and does this mean that some or all of the dogs in the dental treatment groups received some oral care prior to starting the main study?  Or were dogs only handled during this period?  More information is needed on the exact training protocol (handling and treat use) with the dogs during the pre-treatment and treatment periods. 

The dogs was trained to be handled (but no dental treatment) by the veterinary nurse students the week before the treatment started. Added in the text

Also, when were data collected for FAS analysis in the main study?  The text says that dogs were assess each day for this week, but the statistical analysis section indicates that dogs were assessed before and after the study period.  Please clarify.

The FAs was scored everyday and showed in the result section in Figure 7. For the statistics the dogs' individual FAS value were analyzed only between pre and post treatment as described in the statistical section.

L184:  Change 'sweets' to 'treats'.  Here and below.

Changed

L187-188:  What is mean by 'hoarseness with relaxed mouth pores'?

Changed Clarification in the text

L181-202:  Who completed the FAS, and was it completed in real-time or from video?  Were the assessors blind to the study purposes or the treatment groups?

FAS was assessed by the two veterinary nursing students performing the different treatments (tooth brushing and textiles) together during the tooth cleaning each day. Added to the text.

L203-208:  The data that were collected are categorical rather than continuous, so parametric t-tests aren't an appropriate approach for analysis.  Some might argue that parametric stats are ok for the total oral health score, although technically non-parametric approaches are still better since one can't assume that the distance between levels is consistent.  For the other outcomes with only 3-4 levels that are, again, not necessarily consistent between levels, parametric approaches are not a good fit.  The authors are encouraged to explore non-parametric approaches that are better suited to the data collected, and potentially consult with a statistician. Based on the raw data provided it is likely that the overall results will be similar with a non-parametric approach.

All data were analyzed in an Andersson Darling normality test and as the data was normally distributed, the common parametric Student’s test (one-sided paired test) was used.

L211:  Were differences also examined for the control group as scores appear to be numerically lower for the second assessment?

No such comparisons were made. The controls were to confuse the blinded veterinarian and some of them had dental cleaning under anesthesia. Clarified.

Figures 1 & 2:  Replace commas with periods.  e.g., 0.2 rather than 0,2

New figures will be attached

L242-244: In order to discuss comparisons the amount of staining needs to be quantified in some way (scoring system or % tooth covered as measured with Image j or other similar software).  These pictures are striking but are not enough to draw conclusions from on the whole. Please provide specific details for the method of assessment and comparison.

Clarified in the text that the plaque was measured by PI and the staining only for visualization.

L292: Details about rawhide bones access should be included in the Materials and Methods. Did access differ before and after treatment or was it consistently provided both before and after?  If only after it is a serious confound for the study, but if it was also provided before it is unlikely to explain the improvement that occurred.  Please update this section to clarify.

Clarified that the dogs always have free access to the bones.

L327-328:  Based on the data provided no comparisons were made between the textile and toothbrush groups, so you can't really say that there were comparable effects - at most you can say that both groups showed improvements.  However, given that the studies were conducted at the same time you could consider statistically comparing the treatments between groups.

Clarified that all treatments were effective in PI and GI but no comparison was made between treatments.

L339:  You did not present any data on the user experience therefore it isn't appropriate to include comments about this in the discussion.  Instead maybe you could refer to the average dog response which shows that the dogs tolerated the process without biting?

No dogs were biting but was habituating to the treatments.

L358:  Previous studies have looked at habituating animals (including dogs) to different types of handling and procedures, and it would be helpful to incorporate some of this information into this portion of the discussion.

Thank you, new ref are included

L364-365:  Was reliability for scoring FAS assessed before or during the study?

Clarifying how FAS was scored is included

L368-370:  A number of studies have assessed people's ability to detect these subtle signals - please add citations to support this statement.

Thank you, new ref are included

L394-396:  Again, this information is not based on data presented in the paper.  If this information was recorded please include it in the results section.

 The information was not scored as both treatments were included in the FAS scoring for each dog and could therefore not be separated in this study as we believe that the order of the treatment could be influencing the results. We think that the subjective notes from the study still could be of interest to discuss.

Round 2

Reviewer 2 Report

The authors have generally done a nice job of updating the manuscript.  I have a few remaining concerns detailed below.

L138:  It would be helpful to include your brief rationale for the specific treatments (device and paste or lack thereof) either here or at the end of the introduction. Nothing too long, just a quick statement that the devices were used as recommended to see if the standard recommended method that was likely to be used by owners at home was effective.

L149-157 repeats the paragraph above and was likely included in error

L297:  I stand by my original comments about parametric statistics being inappropriate for this data set. Categorical data of this type do not meet the basic assumptions necessary for a parametric approach, even if the Andersson Darling normality test suggests the data are normally distributed.  It is recommended that the authors consult with a statistician.

Table 3, Figures 1 & 2:  Apologies for not noticing previously, but if the only comparison made was pre and post for each treatment without comparing between groups, it is inappropriate to provide a combine pre-treatment level in the graphs.  The data used in the analysis should be presented here (pre for each side of the mouth that is relevant to each treatment presented separately).

Author Response

Thank you for taking time to read and review our work.

Below is our changes point by point

L138:  It would be helpful to include your brief rationale for the specific treatments (device and paste or lack thereof) either here or at the end of the introduction. Nothing too long, just a quick statement that the devices were used as recommended to see if the standard recommended method that was likely to be used by owners at home was effective.

Accepted (Line 127-128)

L149-157 repeats the paragraph above and was likely included in error

Thank you, now corrected.

L297:  I stand by my original comments about parametric statistics being inappropriate for this data set. Categorical data of this type do not meet the basic assumptions necessary for a parametric approach, even if the Andersson Darling normality test suggests the data are normally distributed.  It is recommended that the authors consult with a statistician.

Accepted, now changed to Wilcoxon signed rank test, p-values changed accordingly in the manuscript. (lines 246-262)

Table 3, Figures 1 & 2:  Apologies for not noticing previously, but if the only comparison made was pre and post for each treatment without comparing between groups, it is inappropriate to provide a combine pre-treatment level in the graphs.  The data used in the analysis should be presented here (pre for each side of the mouth that is relevant to each treatment presented separately).

Thank you for pointing out a possible unclarity. The data used in the analysis is the one presented in the table. The mean pre-treatment index score for the toothbrush group in Fig.1 and for the textile group in Fig 2.  The pre-treatment assessment was performed on the whole mouth, as none of the dogs in the study had any differences between their right and left side at the start of the study. This is now clarified on lines 190-193.

We hope that you will find the revised version more clear, and that we have been able to incorporate the suggestions from our reviewers successfully.

Yours respectfully,

Round 3

Reviewer 2 Report

All suggested changes have been made and I have no further concerns.